# Transient acidosis while retrieving a fear-related memory enhances its lability

Jianyang Du[1,2]*, Margaret P Price[1], Rebecca J Taugher[3], Daniel Grigsby[1], Jamison J Ash[1], Austin C Stark[1], Md Zubayer Hossain Saad[2], Kritika Singh[2], Juthika Mandal[2], John A Wemmie[3,4,5], Michael J Welsh[1,4,5,6]*

[1]Departments of Internal Medicine, Roy J and Lucille A Carver College of Medicine, University of Iowa, Iowa City, United States; [2]Department of Biological Sciences, University of Toledo, Toledo, United States; [3]Department of Psychiatry, Roy J and Lucille A Carver College of Medicine, University of Iowa, Iowa City, United States; [4]Department of Molecular Physiology and Biophysics, Roy J and Lucille A Carver College of Medicine, University of Iowa, Iowa City, United States; [5]Department of Neurosurgery, Roy J and Lucille A Carver College of Medicine, University of Iowa, Iowa City, United States; [6]Howard Hughes Medical Institute, Roy J and Lucille A Carver College of Medicine, University of Iowa, Iowa City, United States

**Abstract** Attenuating the strength of fearful memories could benefit people disabled by memories of past trauma. Pavlovian conditioning experiments indicate that a retrieval cue can return a conditioned aversive memory to a labile state. However, means to enhance retrieval and render a memory more labile are unknown. We hypothesized that augmenting synaptic signaling during retrieval would increase memory lability. To enhance synaptic transmission, mice inhaled $CO_2$ to induce an acidosis and activate acid sensing ion channels. Transient acidification increased the retrieval-induced lability of an aversive memory. The labile memory could then be weakened by an extinction protocol or strengthened by reconditioning. Coupling $CO_2$ inhalation to retrieval increased activation of amygdala neurons bearing the memory trace and increased the synaptic exchange from $Ca^{2+}$-impermeable to $Ca^{2+}$-permeable AMPA receptors. The results suggest that transient acidosis during retrieval renders the memory of an aversive event more labile and suggest a strategy to modify debilitating memories.

*For correspondence: jianyang. du@utoledo.edu (JD); michael-welsh@uiowa.edu (MJW)

**Competing interests:** The authors declare that no competing interests exist.

## Introduction

Fear memories that are excessive or inappropriately triggered can be disabling in disorders such as post-traumatic stress disorder and phobias (*Izquierdo et al., 2016*; *Parsons and Ressler, 2013*). These illnesses remain a frequent cause of morbidity and mortality, and better treatments are sorely needed. A common behavioral treatment has been extinction training (exposure therapy), in which the threatening cue is repeatedly presented without the feared negative consequences (*Izquierdo et al., 2016*; *Parsons and Ressler, 2013*). However, extinction training does not erase the original memory, and thus it can reemerge with time, stress, and contextual cues.

Potentially more effective treatments might change the original memory so that the threatening stimulus is less strongly associated with the harmful event. An opportunity to do this arose with the realization that following acquisition and initial consolidation of a memory, the memory can be returned to a labile state (*Nader et al., 2000*; *Kim et al., 2010*; *Monfils et al., 2009*; *Clem and Huganir, 2010*; *Hong et al., 2013*). While in the labile state it can be modified. This process has often been studied in rodents with Pavlovian aversive conditioning, also called fear conditioning. In

commonly used auditory conditioning paradigms, a tone (conditioned stimulus) is paired with a foot shock (unconditioned stimulus), and at a later time freezing is measured when the tone is presented (conditioned response). If the memory is retrieved, for example by playing a single tone one day after aversive conditioning, the memory returns to a labile state. During the <6 hr period of lability, the memory can be modified (updated). For example, presenting an extinction protocol during the labile state can reduce the conditioned freezing response in rats and mice (*Nader et al., 2000*; *Kim et al., 2010*; *Monfils et al., 2009*; *Clem and Huganir, 2010*; *Hong et al., 2013*). Studies in humans suggest a similar process (*Schiller et al., 2010*; *Agren et al., 2012*; *Liu et al., 2014*; *Björkstrand et al., 2016*). Nevertheless, the response can be variable, depends on multiple factors, and is not always observed in rodents (*Chan et al., 2010*; *Costanzi et al., 2011*) or humans (*Klucken et al., 2016*; *Golkar et al., 2012*). Thus, further increasing lability might make the memory more amenable to modification. However, the means to enhance lability are unknown.

Lateral amygdala neurons and glutamatergic transmission play a central role in the acquisition and retrieval of aversive memories (*Maren, 2005*; *Phelps and LeDoux, 2005*). However, the molecular mechanisms by which retrieval causes a consolidated memory to become labile are not fully understood. With aversive conditioning, synaptic strength increases, and $Ca^{2+}$-impermeable α-amino-3-hydroxyl-5-methyl- 4-isoxazole-propionate (AMPA) receptors (CI-AMPARs) increase on lateral amygdala synapses (*Clem and Huganir, 2010*; *Hong et al., 2013*). Insight into the generation of the labile state came with the finding that retrieval initiates a swap from CI-AMPARs to $Ca^{2+}$-permeable AMPA receptors (CP-AMPARs) (*Clem and Huganir, 2010*; *Hong et al., 2013*). However, the increase in CP-AMPARs is transient, and they are replaced by CI-AMPARs over a time course approximately the same as the period of memory lability. The switch to increased CP-AMPARs is a key molecular event that supports and is required for retrieval to render a memory temporarily labile.

In addition to glutamate receptors, lateral amygdala neurons also express acid sensing ion channels (ASICs) (*Wemmie et al., 2003*; *Ziemann et al., 2009*). ASICs are non-voltage gated $Na^+$- and $Ca^{2+}$-permeable channels that are activated by extracellular acidosis (*Deval et al., 2010*; *Sherwood et al., 2012*). Previous studies identified protons as a neurotransmitter and ASICs as their postsynaptic receptor in lateral amygdala neurons (*Du et al., 2014*; *Zha et al., 2006*). Several observations suggest that proton:ASIC signaling modulates synaptic activity rather than serving as the major neurotransmitter-receptor pair. Proton:ASIC signaling has little effect on neurotransmission under basal conditions (*Du et al., 2014*; *Wemmie et al., 2002*). ASIC excitatory post-synaptic currents (EPSCs) are small compared to glutamate-dependent EPSCs (*Du et al., 2014*). Moreover, extracellular acidification is not sufficient to induce long-term potentiation on its own; induction requires delivery of test pulses to the presynaptic neuron (*Du et al., 2014*). Yet, this signaling system is required for normal synaptic plasticity (*Du et al., 2014*; *Wemmie et al., 2002*; *Chiang et al., 2015*). Protons may be co-released into synapses with glutamate (*Du et al., 2014*; *DeVries, 2001*; *Palmer et al., 2003*), and proton:ASIC signaling may be particularly important during intense presynaptic stimulation, which causes the greatest reductions in extracellular pH (*Du et al., 2014*; *MacLean and Jayaraman, 2016*). Importantly, ASICs are abundantly expressed in the amygdala and are required for normal amygdala-dependent memory (*Ziemann et al., 2009*; *Chiang et al., 2015*; *Coryell et al., 2008*).

If aversive memories are to be modified, they must first return to a labile state. We hypothesized that enhancing synaptic signaling while presenting a retrieval cue would increase the lability of an aversive memory. Then, subsequent interventions could more effectively modify the memory. Because ASIC channel activity can enhance synaptic signaling and because ASICs are abundant in amygdala neurons, we tested the hypothesis that transient acidification during retrieval would increase memory lability.

## Results

### Inhaling $CO_2$ enhances the ability of retrieval to render an aversive memory labile

Previous studies described auditory conditioning protocols in which a retrieval event followed by extinction could attenuate a fear-related memory (*Monfils et al., 2009*; *Clem and Huganir, 2010*;

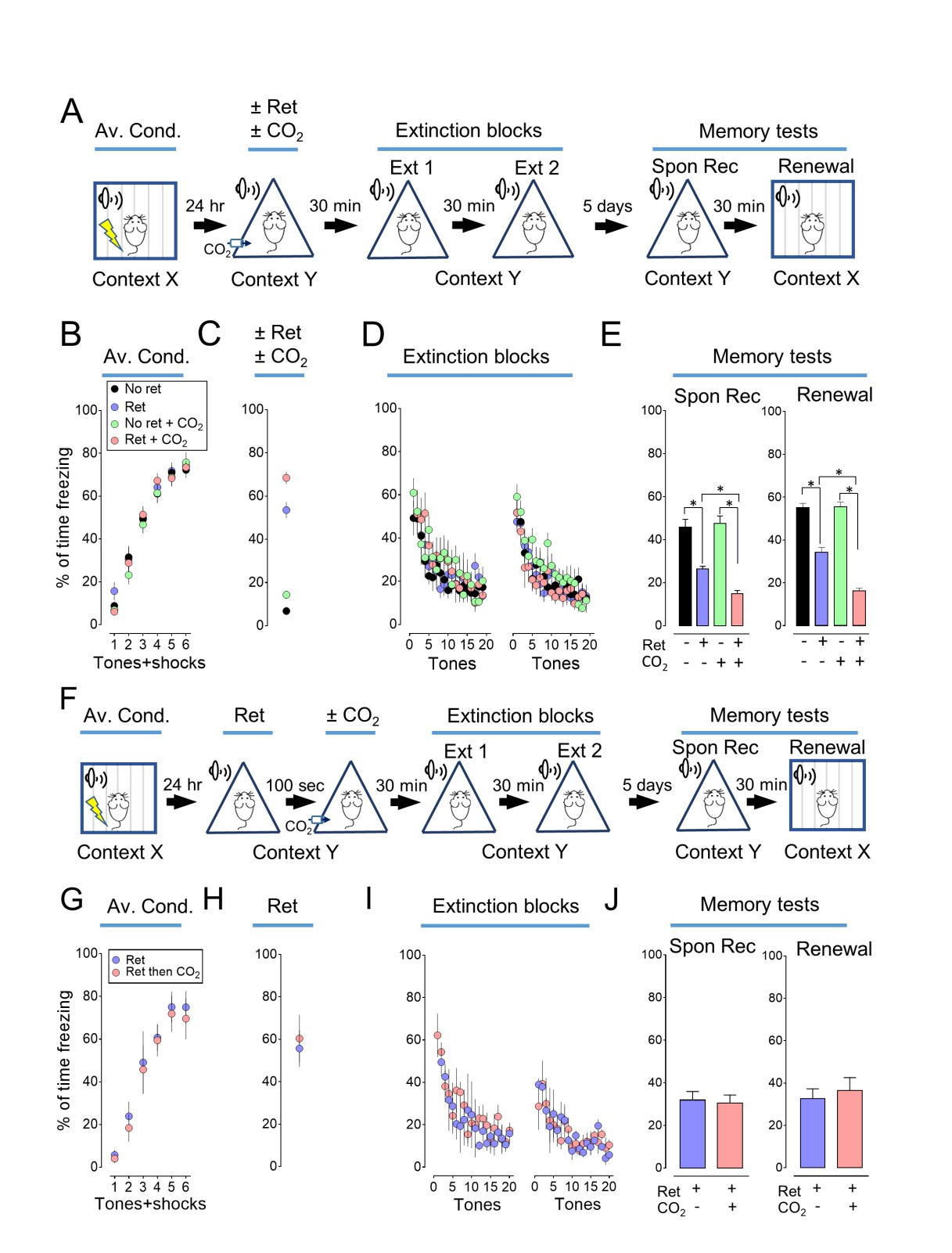

**Figure 1.** Inhaling CO₂ during a retrieval tone augments the effect of extinction. (**A**) Schematic of protocol for auditory aversive conditioning (also called fear conditioning), memory modification, and testing. The protocol contained four main components. (1) On day 1, mice underwent aversive conditioning (Av. Cond.) during which they received 6 tones paired with 6 foot shocks in context X. (2) On day 2, mice were placed in a modified chamber (context Y) and were randomly assigned to one of four groups. After 5 min, one group was presented with a single tone (Ret) and then

*Figure 1 continued on next page*

*Figure 1 continued*

remained in the chamber for two more minutes. One group breathed 10% $CO_2$ for 7 min. One group breathed 10% $CO_2$ and was presented with the tone. And one group received neither $CO_2$ nor retrieval. (3) Thirty minutes later in context Y, all mice were treated with two blocks of extinction protocols (Ext 1 and Ext 2), each consisting of 20 tones; mice in the retrieval groups received 19 instead of 20 tones in the first block of extinction tones. (4) On day 7, mice were tested for freezing in response to 4 tones delivered in context Y (Spontaneous Recovery). Thirty minutes later they were tested for freezing in response to 4 tones delivered in context X (Renewal). (B–E) Data are the percentage of time mice were freezing in response to tones during aversive conditioning (B), the Ret/$CO_2$ intervention (C), extinction (D), and during the two memory tests (E). (F) Schematic of protocol for auditory aversive conditioning, memory modification, and testing. The protocol was identical to that shown in panel A with the exception that the retrieval tone was presented and then 100 s later mice inhaled 10% $CO_2$ for 7 min. (G–J) Data are the percentage of time mice were freezing during aversive conditioning (G), retrieval (H), extinction (I), and the two memory tests (J). Data are mean±SEM. n = 16 mice in each group for panels B-E and n = 8 mice in each group for panels G-J. * indicates $p<0.05$ by one-way ANOVA with Tukey's post hoc multiple comparison for panels B-E. For panel E, No ret vs Ret, $p=0.0001$; No ret vs No ret + $CO_2$, $p=0.9683$; No ret vs Ret + $CO_2$, $p<0.0001$; Ret vs No ret + $CO_2$, $p<0.0001$; Ret vs Ret + $CO_2$, $p=0.0300$; No ret + $CO_2$ vs Ret + $CO_2$, $p<0.0001$. For panel E right, No ret vs Ret, $p<0.0001$; No ret vs No ret + $CO_2$, $p=0.9974$; No ret vs Ret + $CO_2$, $p<0.0001$; Ret vs No ret + $CO_2$, $p<0.0001$; Ret vs Ret + $CO_2$, $p<0.0001$; No ret + $CO_2$ vs Ret + $CO_2$, $p<0.0001$. There were no statistically significant differences between groups in panels G-J by unpaired Student's *t*-test.

The following source data and figure supplements are available for figure 1:

**Source data 1.** Contexts for experiments.
**Figure supplement 1.** Inhaling $CO_2$ during a retrieval tone augments the effect of extinction.
**Figure supplement 2.** The effect of $CO_2$ on memory retrieval is concentration dependent.

*Hong et al., 2013*). We used a similar paradigm (*Figure 1A*, *Figure 1—source data 1*). On day 1, mice were trained to associate 6 tones with 6 foot shocks in context X (*Figure 1B*). On day 2, the mice were placed in a different context (context Y) and a single tone (retrieval event) was presented (*Figure 1C*). Thirty minutes later, all mice underwent an extinction protocol with two blocks of 20 tones in context Y (total of 39 tones for mice exposed to the retrieval tone) (*Figure 1D*). By the end of the extinction protocol, freezing fell to a low level. On day 7, mice were tested for spontaneous recovery by measuring freezing in response to tones in context Y (*Figure 1E*). Then, 30 min later, they were tested for renewal by measuring the response to tones while they were in context X (*Figure 1E*). Consistent with earlier reports (*Monfils et al., 2009*; *Clem and Huganir, 2010*; *Hong et al., 2013*), retrieving the memory with a single tone before extinction reduced freezing in both spontaneous recovery and renewal tests.

We tested the hypothesis that presenting the retrieval tone while the brain was transiently acidic would increase memory lability and hence the efficacy of extinction in attenuating the freezing response. To acidify the brain, mice inhaled 10% $CO_2$. We used $CO_2$ inhalation because it rapidly acidifies brain pH including in the lateral amygdala (*Ziemann et al., 2009*), the effect reverses quickly when $CO_2$ is stopped (*Ziemann et al., 2009*), and it can be used safely in humans (*Poma et al., 2005*; *Bailey et al., 2005*). Mice inhaled 10% $CO_2$ for 5 min before and then during presentation of the single retrieval tone for a total of 7 min (*Figure 1A*). When the retrieval tone and $CO_2$ inhalation were paired, conditioned freezing during the memory tests was reduced compared to retrieval by itself (*Figure 1E*, *Figure 1—figure supplements 1* and *2*). In contrast, inhaling 10% $CO_2$ by itself did not alter the memory tests, suggesting that the effect of $CO_2$ was to enhance the effects of retrieval.

We also tested the effect of separating the retrieval tone and $CO_2$ inhalation in time (*Figure 1F*). When $CO_2$ inhalation began 100 s after the retrieval tone and continued for 7 min, the conditioned freezing response during the memory tests was not reduced compared to retrieval by itself (*Figure 1G–1J*).

Modification of transiently labile memories has most often involved reductions in memory strength, for example with extinction training (*Monfils et al., 2009*; *Clem and Huganir, 2010*; *Hong et al., 2013*). However, labile memories may also be strengthened (*Fukushima et al., 2014*). We reasoned that if inhaling $CO_2$ during retrieval made a memory more labile, then it might also be more susceptible to strengthening by reconditioning. We used distinct contexts to avoid contextual effects (*Figure 1—source data 1*). On day 1, mice were trained with 3 tones paired with foot shocks

in context X (*Figure 2A and B*). On day 2, mice were randomly assigned to one of four groups, receiving a single retrieval tone or not, with or without 10% $CO_2$ in context Y. Thirty minutes after retrieval, all mice received an additional tone paired with a foot shock in context Z to recondition the memory. On day 3, we tested the aversive memory in context ZZ and found that a prior retrieval tone enhanced reconditioning (*Figure 2C*, *Figure 2—figure supplements 1* and *2*). Moreover, retrieving the memory during $CO_2$ inhalation further enhanced reconditioning. In contrast, inhaling $CO_2$ by itself had no effect on subsequent reconditioning.

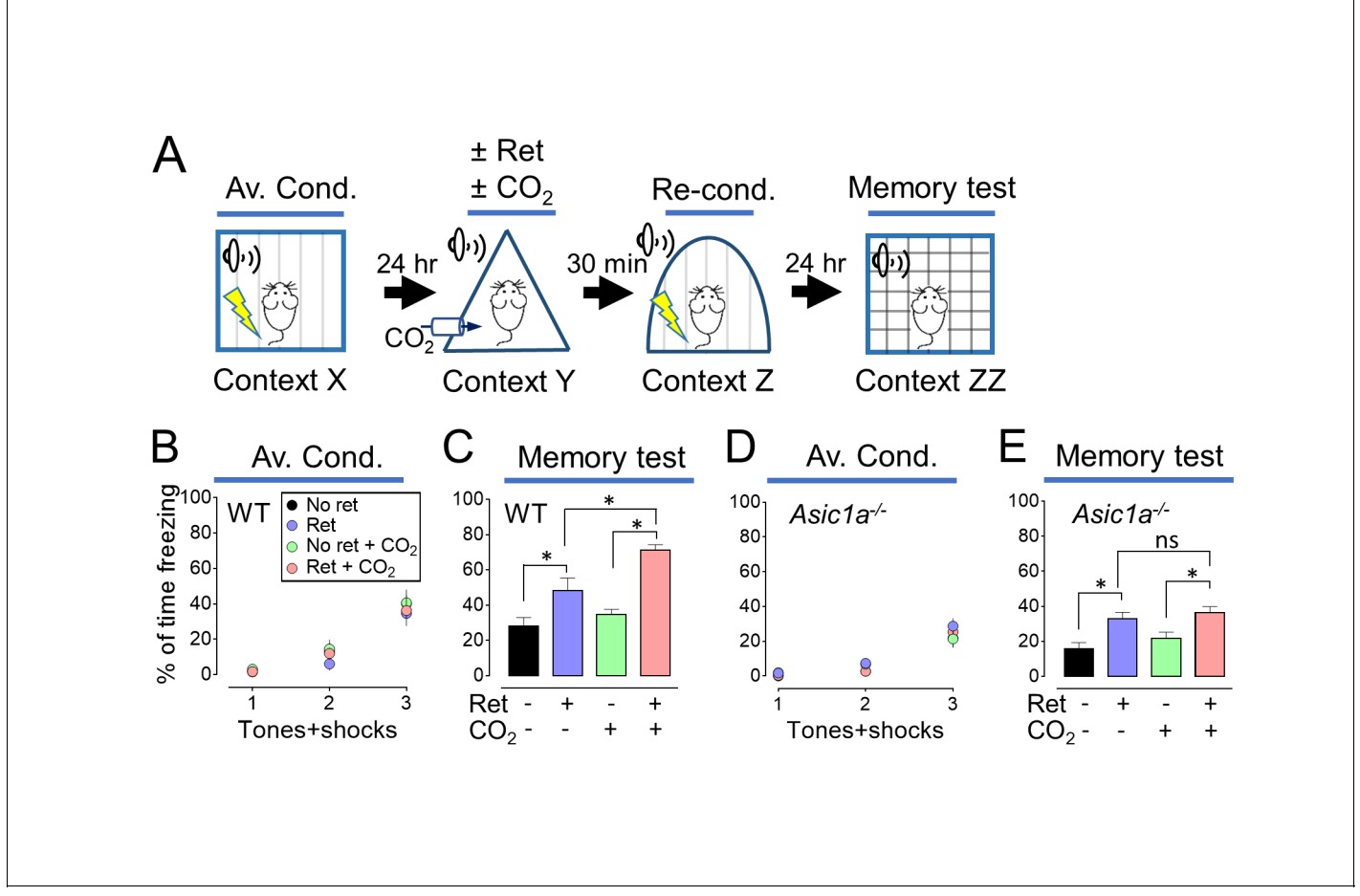

**Figure 2.** Inhaling $CO_2$ during a retrieval tone augments the effect of reconditioning. (**A**) Schematic of protocol for memory enhancement. During aversive conditioning, mice received 3 paired tones and foot shocks in context X. The Ret/$CO_2$ period in context Y was as described for *Figure 1A*. Thirty minutes later, mice received a tone paired with a foot shock in context Z as reconditioning (Re-Cond.). Memories were tested 1 day later by presenting 4 tones in context ZZ. See also *Figure 1—source data 1* and *Figure 2—figure supplements 1* and *2*. (**B,C**) Percentage of time freezing during aversive conditioning (**B**) and the memory testing (**C**) in wild-type (WT) mice. Data are mean±SEM. n = 16 mice in each group. * indicates p<0.05 by ANOVA with Tukey's post hoc multiple comparison. No ret vs Ret, p=0.0295; No ret vs No ret + $CO_2$, p=0.7774; No ret vs Ret + $CO_2$, p<0.0001; Ret vs No ret + $CO_2$, p=0.1763; Ret vs Ret + $CO_2$, p=0.0055; No ret + $CO_2$ vs Ret + $CO_2$, p<0.0001. (**D,E**) Data as in panels B and C except in *Asic1a*$^{-/-}$ mice. Data are mean±SEM. n = 16 mice in each group. * indicates p<0.05 by ANOVA with Tukey's post hoc multiple comparison. 'ns' indicates not statistically significant. No ret vs Ret, p=0.0055; No ret vs No ret + $CO_2$, p=0.6228; No ret vs Ret + $CO_2$, p=0.0006; Ret vs No ret + $CO_2$, p=0.1242; Ret vs Ret + $CO_2$, p=0.9049; No ret + $CO_2$ vs Ret + $CO_2$, p=0.0248.

The following figure supplements are available for figure 2:

**Figure supplement 1.** Distinct contexts were used to test the effect of retrieval on memory enhancement.

**Figure supplement 2.** Reconditioning enhances an aversive memory.

These data suggest that when mice inhale $CO_2$ while a retrieval cue is presented, the memory becomes more labile and it can then either be weakened by extinction or strengthened by reconditioning.

## $CO_2$ enhancement of retrieval-induced lability requires ASICs

Inhaling 10% $CO_2$ reduces brain pH and activates ASIC channels (*Ziemann et al., 2009*). To test if ASIC currents are required for $CO_2$ to enhance the effect of retrieval, we studied $Asic1a^{-/-}$ mice; disrupting the $Asic1a$ gene eliminates neuronal ASIC currents in response to acidic challenges with pH >5 (*Wemmie et al., 2002*). In $Asic1a^{-/-}$ mice, inhaling $CO_2$ during retrieval failed to further increase the conditioned response (*Figure 2D and E*). The requirement for ASIC channels, which are proton-activated, suggests that $CO_2$ exerts its effect by acidifying pH.

## $CO_2$ inhalation during retrieval augments the exchange of AMPA receptors

Previous studies revealed that retrieval induces a rapid and transient exchange from CI-AMPARs to CP-AMPARs at lateral amygdala synapses (*Clem and Huganir, 2010*; *Hong et al., 2013*). CI-AMPARs contain GluA2 subunits, whereas CP-AMPARs lack GluA2, but contain GluA1 subunits (*Isaac et al., 2007*). Ten minutes after retrieval, we prepared brain slices and stimulated thalamic inputs to the lateral amygdala. We assayed pyramidal neurons for AMPAR current rectification, a signature of CP-AMPARs. Consistent with earlier reports (*Clem and Huganir, 2010*; *Hong et al., 2013*), retrieval increased rectification (*Figure 3A and B*). $CO_2$ alone had no effect. However, inhaling $CO_2$ while the retrieval tone was presented caused more rectification than retrieval alone. In contrast, adding $CO_2$ to retrieval in $Asic1a^{-/-}$ mice failed to increase rectification (*Figure 3C*).

We also applied an inhibitor of CP-AMPARs, 1-naphthylacetyl spermine (NASPM). Compared to controls, retrieval increased NASPM-sensitive EPSCs (*Figure 3D*). Inhaling $CO_2$ during retrieval further increased NASPM-sensitive EPSCs. In contrast, in $Asic1a^{-/-}$ mice, the NASPM-inhibited current did not differ between retrieval alone and retrieval plus $CO_2$ (*Figure 3E*).

To test the possibility that retrieval while inhaling $CO_2$ reduces synaptic strength, we exposed aversive conditioned mice to retrieval or not, with or without $CO_2$ inhalation. Ten minutes later, we prepared brain slices, stimulated thalamic inputs, and measured the ratio of AMPAR-EPSCs to N-methyl-D-aspartate receptor (NMDAR) EPSCs in lateral amygdala neurons. Delivering 10% $CO_2$ during retrieval had little effect on the AMPAR/NMDAR-EPSC ratio (*Figure 3F*). Moreover, the amplitude, frequency, and decay time of miniature EPSCs (mEPSCs) were not altered (*Figure 3—figure supplement 1*). These data suggest that $CO_2$ inhalation plus retrieval did not change synaptic strength, consistent with previous data that retrieval alone did not alter synaptic strength (*Hong et al., 2013*).

A shift from CI-AMPARs to CP-AMPARs without an effect on synaptic strength indicates endocytosis of CI-AMPAR (*Hong et al., 2013*; *Shepherd and Huganir, 2007*; *Santos et al., 2009*). To inhibit CI-AMPAR endocytosis, the day after aversive conditioning, we injected Tat-GluA2$_{3Y}$ or Tat-GluA2$_{3A}$ peptides into the mouse lateral amygdala. Tat-GluA2$_{3Y}$ is a cell-permeable peptide that mimics the C-terminal tail of GluA2 and prevents GluA2-dependent AMPAR endocytosis (*Hong et al., 2013*; *Brebner et al., 2005*). Tat-GluA2$_{3A}$ served as a control. One hour after injection, we presented a retrieval tone and 10% $CO_2$. We found that 30 min later, Tat-GluA2$_{3Y}$ reduced AMPAR current rectification compared to injection of a control peptide, Tat-GluA2$_{3A}$ (*Figure 3G*). These results predicted that inhibiting AMPAR exchange would prevent $CO_2$-enhancement of retrieval. After aversive conditioning, we injected Tat-GluA2$_{3Y}$ or Tat-GluA2$_{3A}$ into the mouse lateral amygdala bilaterally (*Cazakoff and Howland, 2011*; *Dias et al., 2012*). One hour later, we presented a retrieval tone with or without 10% $CO_2$, followed by the extinction protocol (*Figure 3—figure supplement 2*). Compared to the Tat-GluA2$_{3A}$ control, Tat-GluA2$_{3Y}$ prevented the effect of $CO_2$ on conditioned freezing in the spontaneous recovery and renewal memory tests (*Figure 3—figure supplement 2H and I*).

These data suggest that inhaling $CO_2$ during presentation of a single retrieval tone increases the exchange from CI-AMPARs to CP-AMPARs at lateral amygdala synapses. This exchange is critical for inducing lability of an aversive memory (*Clem and Huganir, 2010*; *Hong et al., 2013*). Because

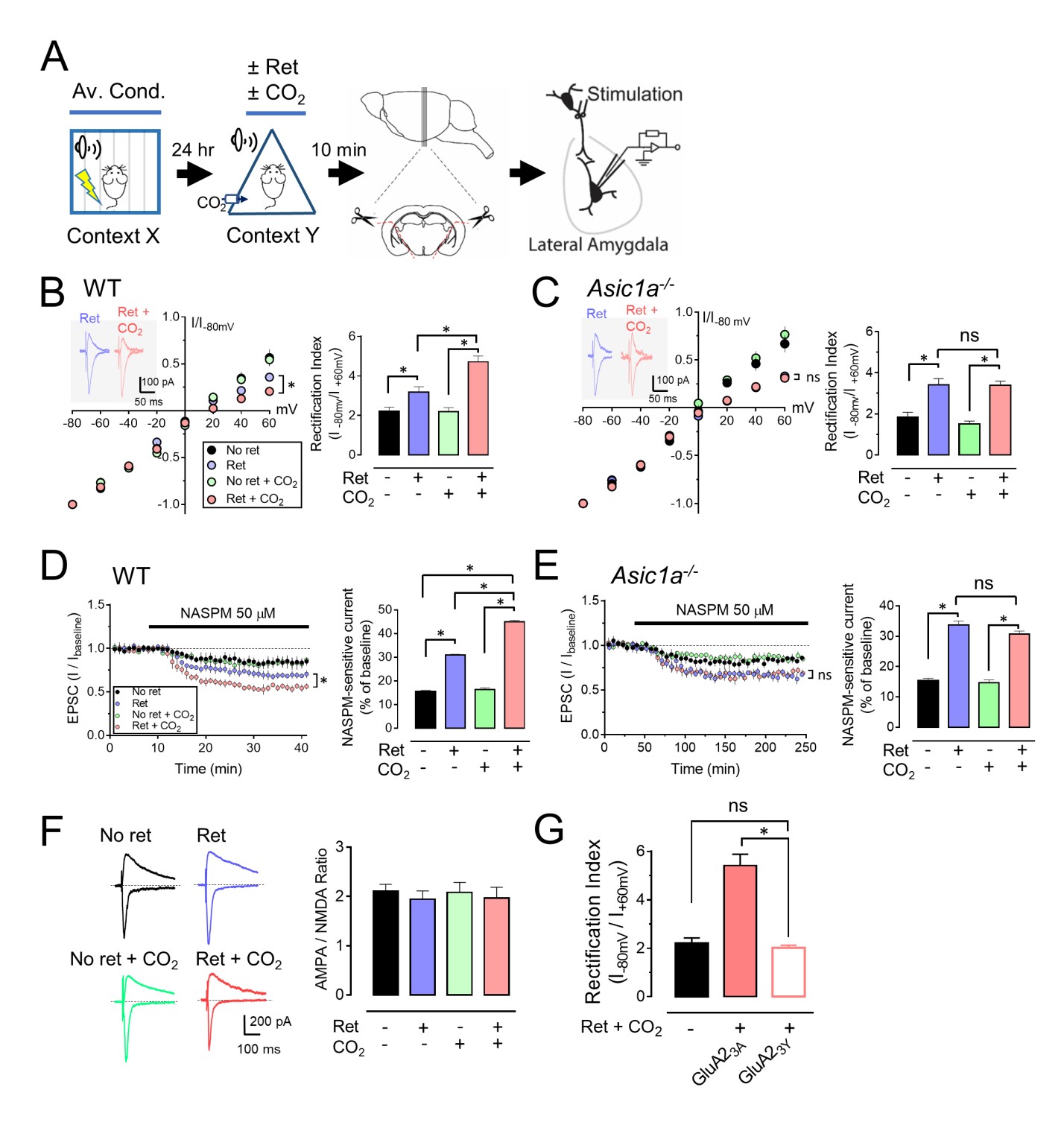

**Figure 3.** $CO_2$ inhalation enhances retrieval-dependent AMPAR current rectification. (**A**) Schematic of experimental procedure. Aversive conditioning and Ret/$CO_2$ were as described for *Figure 1A*. Ten minutes later, brain slices were prepared, and patch-clamp recordings were made from lateral amygdala pyramidal neurons. Stimulation was of thalamic inputs. (**B,C**) Left, AMPAR current-voltage relationships in lateral amygdala pyramidal neurons. Insets show example of current from a mouse treated with retrieval and a mouse treated with retrieval plus $CO_2$. Right, AMPAR rectification index ($I_{-80}$ mV / $I_{+60}$ mV). Data are from wild-type (**B**) and *Asic1a$^{-/-}$* (**C**) mice. (**D,E**) Left, time-course showing effect of NASPM (blocks CP-AMPAR activity) application on EPSCs. D-APV (100 μM) was present throughout. Right, amplitude of NASPM-inhibited EPSCs as a percentage of current before NASPM addition. Data are from wild-type (**D**) and *Asic1a$^{-/-}$* (**E**) mice. (**B–E**) Data are mean±SEM. n = 20–24 for each group. * indicates $p < 0.05$ by

*Figure 3 continued on next page*

*Figure 3 continued*

ANOVA with Tukey's post hoc multiple comparison. 'ns' indicates not statistically significant. In B, No ret vs Ret, p=0.0281; No ret vs No ret + $CO_2$, p>0.9999; No ret vs Ret + $CO_2$, p<0.0001; Ret vs No ret + $CO_2$, p=0.0297; Ret vs Ret + $CO_2$, p=0.0005; No ret + $CO_2$ vs Ret + $CO_2$, p<0.0001. In C, No ret vs Ret, p<0.0001; No ret vs No ret + $CO_2$, p=0.7355; No ret vs Ret + $CO_2$, p<0.0001; Ret vs No ret + $CO_2$, p<0.0001; Ret vs Ret + $CO_2$, p>0.9999; No ret + $CO_2$ vs Ret + $CO_2$, p<0.0001. In D, No ret vs Ret, p<0.0001; No ret vs No ret + $CO_2$, p=0.8939; No ret vs Ret + $CO_2$, p<0.0001; Ret vs No ret + $CO_2$, p<0.0001; Ret vs Ret + $CO_2$, p<0.0001; No ret + $CO_2$ vs Ret + $CO_2$, p<0.0001. In E, No ret vs Ret, p<0.0001; No ret vs No ret + $CO_2$, p=0.9452; No ret vs Ret + $CO_2$, p<0.0001; Ret vs No ret + $CO_2$, p<0.0001; Ret vs Ret + $CO_2$, p=0.1947; No ret + $CO_2$ vs Ret + $CO_2$, p<0.0001. (F) Left, examples of excitatory postsynaptic currents (EPSCs) recorded at −80 mV (AMPAR-EPSCs) and at +60 mV (NMDAR-EPSCs). Right, AMPAR/NMDAR EPSC ratios. Current amplitudes were measured 70 ms after onset. n = 12–16 for each group. (G) Rectification index of AMPAR-EPSCs. Aversive conditioning was done as in *Figure 1A*. Twenty-four hours later, mice received a microinjection of GluA2$_{3Y}$ or GluA2$_{3A}$ (control) into the amygdala. One hour later, mice were presented with a retrieval tone while inhaling 10% $CO_2$. Ten minutes later, brain slices were prepared, and AMPAR-EPSCs were recorded from lateral amygdala pyramidal neurons. Data are mean±SEM of rectification index of AMPA-EPSCs. n = 24–40 for each group. * indicates p<0.05 by ANOVA with Tukey's post hoc multiple comparison. 'ns' indicates not statistically significant. No ret vs GluA2$_{3A}$, p<0.0001; No ret vs GluA2$_{3Y}$, p=0.9197; GluA2$_{3A}$ vs GluA2$_{3Y}$, p<0.0001.

The following figure supplements are available for figure 3:

**Figure supplement 1.** Inhaling $CO_2$ does not alter synaptic strength in lateral amygdala neurons.

**Figure supplement 2.** Prevention of shift from CI-AMPARs to CP-AMPARs weakens $CO_2$ induced lability.

disrupting ASIC currents prevented these effects of $CO_2$, the data suggest that the effect of $CO_2$ is mediated by acidosis.

## Injecting acid into the amygdala mimics the effect of $CO_2$ inhalation on AMPAR current

$CO_2$ inhalation will cause acidosis throughout the brain. Therefore, we asked whether reducing pH specifically in the amygdala would affect AMPAR exchange. Five minutes before delivering a retrieval tone, we microinjected acidic saline into the amygdala (*Figure 4A*), which reduces pH to ~6.8 assayed with a micro pH sensor (*Ziemann et al., 2009*); note that inhaling 10% $CO_2$ reduces pH to ~6.9 (19). We removed the amygdala 10 min after the tone and recorded AMPAR-EPSCs. The combination of retrieval plus acid injection produced greater AMPAR current rectification than retrieval alone (*Figure 4B and C*). Notably, in *Asic1a*$^{-/-}$ mice, acid microinjection paired with retrieval did not potentiate AMPAR current rectification. These results suggest that acidifying the amygdala activates ASIC channels, which when paired with retrieval renders an aversive memory more labile.

## $CO_2$ inhalation does not augment AMPA receptor exchange at synapses between cortex and lateral amygdala

When not delivered during retrieval, inhaling $CO_2$ and/or acid injection failed to alter behavioral responses to extinction and reconditioning or the exchange of AMPA receptors. Those results suggested that acidosis primarily affected synapses that were being activated by the retrieval tone. In the studies described above, we assayed synapses between the thalamus and lateral amygdala; aversive conditioning strengthens those synapses in vivo and in vitro (*Rumpel et al., 2005*; *Tsvetkov et al., 2002*). Auditory aversive conditioning also involves synapses between the cortex and lateral amygdala, although the specific contributions of the two inputs remain uncertain (*Shin, 2012*).

When we stimulated cortical input to the lateral amygdala, retrieval failed to potentiate AMPAR current rectification (*Figure 5A*). That negative result provided an opportunity to test the specificity of $CO_2$ inhalation. Inhaling $CO_2$ did not enhance AMPAR current rectification when delivered during the retrieval tone (*Figure 5A*). There was also no enhancement in *Asic1a*$^{-/-}$ mice (*Figure 5B*). Thus, although inhaling $CO_2$ reduces pH throughout the brain, the effect on AMPA receptor exchange was limited to synapses that were being rendered labile by a retrieval tone.

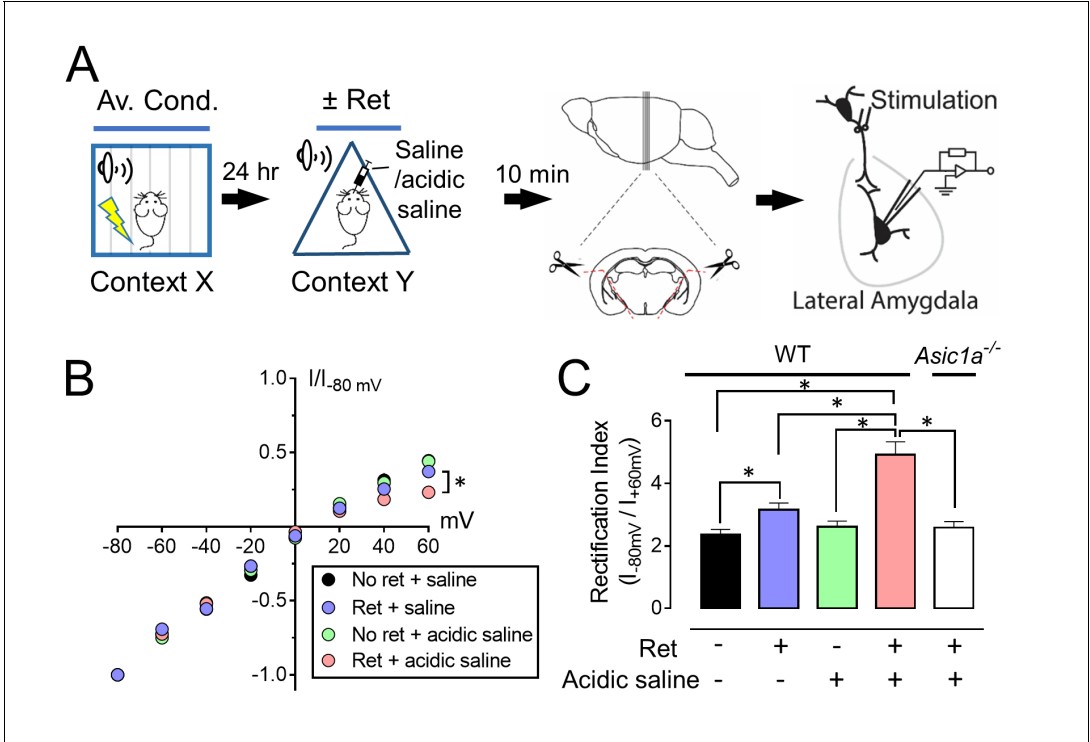

**Figure 4.** Injecting acid into the amygdala before retrieval enhances exchange of AMPARs. (**A**) Schematic of experimental procedure. Aversive conditioning was done as in *Figure 1A*. On day 2, mice received a microinjection of acidic saline (to reduce pH to ~6.8) or saline (pH ~7.35) into the amygdala. Five minutes later, they were presented with a single tone or not. Ten minutes after that, brain slices were prepared and AMPAR-EPSCs were recorded. (**B**) AMPAR current-voltage relationships in lateral amygdala pyramidal neurons. (**C**) Rectification index of AMPA-EPSCs. Data are mean±SEM. n = 18–24 for each group. * indicates $p < 0.05$ by ANOVA with Tukey's post hoc multiple comparison. No ret + saline vs Ret + saline, p=0.0431; No ret vs No ret + acidic saline, p=0.9781; No ret vs Ret + acidic saline, p<0.0001; No ret (WT) vs Ret + acidic saline ($Asic1a^{-/-}$), p=0.9851; Ret + saline vs No ret + acidic saline, p=0.6081; Ret + saline vs Ret + acidic saline, p=0.0002; Ret + saline (WT) vs Ret + acidic saline ($Asic1a^{-/-}$), p=0.5596; No ret + acidic saline vs Ret + acidic saline, p<0.0001; No ret + acidic saline (WT) vs Ret + acidic saline ($Asic1a^{-/-}$), p>0.9999; Ret + acidic saline (WT) vs Ret + acidic saline ($Asic1a^{-/-}$), p<0.0001.

## $CO_2$ enhances activation of amygdala neurons

The importance of the intracellular $Ca^{2+}$ concentration, $[Ca^{2+}]_i$, for neuronal activity suggested that increasing $CO_2$ by 10% might enhance a stimulation-induced rise in $[Ca^{2+}]_i$. Activation of ASICs can increase $[Ca^{2+}]_i$ by conducting $Ca^{2+}$ and/or by enhancing membrane depolarization and thereby the activity of voltage-sensitive $Ca^{2+}$ channels (*Deval et al., 2010*; *Sherwood et al., 2012*; *Zha et al., 2006*; *Xiong et al., 2004*). To assay $[Ca^{2+}]_i$, we loaded a fluorescent $Ca^{2+}$ indicator (Oregon Green 488 BAPTA-6F) into amygdala neurons and stimulated thalamic inputs to induce postsynaptic $Ca^{2+}$ influx (*Stosiek et al., 2003*) (*Figure 6A*). Stimuli of increasing frequency progressively increased the $[Ca^{2+}]_i$ signal (*Figure 6B and C*). When we raised the perfusate $CO_2$ concentration from the control of 5% (pH 7.35) to 15% (pH 6.8), presynaptic stimulation generated a greater increase in the $[Ca^{2+}]_i$ signal. The changes were attenuated in $Asic1a^{-/-}$ mice (*Figure 6D–6F*). These results suggest that the acidic pH produced by $CO_2$ enhances stimulation-dependent increases in $[Ca^{2+}]_i$.

Increased neuronal activity can increase phosphorylation of CREB at residue Ser133 (*Sheng et al., 1991*). Previous work suggested that such phosphorylation is required for $Ca^{2+}$-induced-*Fos* transcription and for reconsolidation of retrieved fear memories (*Sheng et al., 1990*; *Kida et al., 2002*). We found that inhaling $CO_2$ during a retrieval tone increased phosphorylation of CREB Ser133 compared to retrieval alone or $CO_2$ alone (*Figure 6G*).

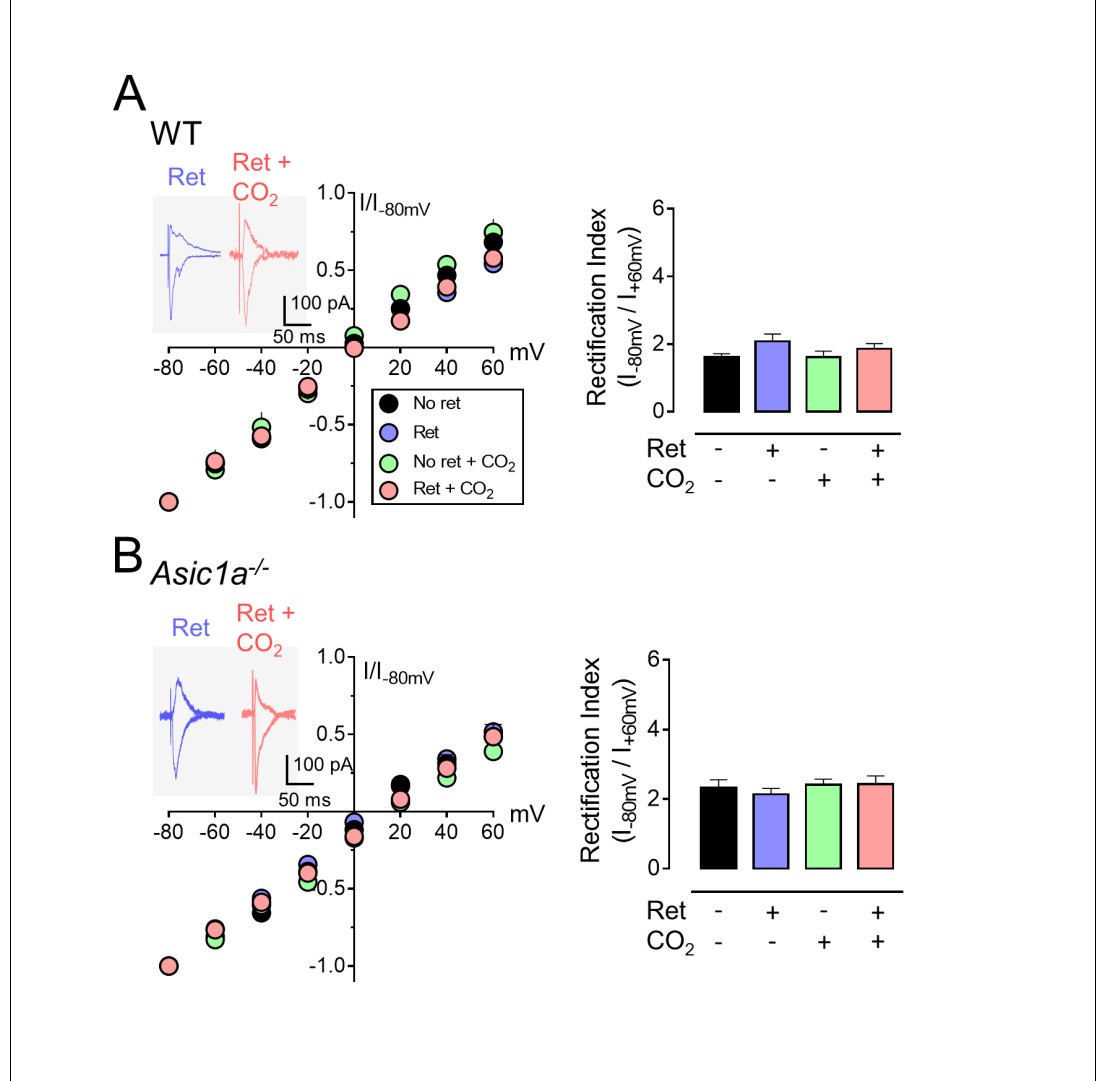

**Figure 5.** Inhaling $CO_2$ does not enhance AMPAR exchange at synapses between the cortex and lateral amygdala. The procedure was the same as that shown in *Figure 3A* except that stimulation was of cortical inputs. (**A,B**) Left, AMPAR current-voltage relationships in lateral amygdala pyramidal neurons. Insets show example of current from a mouse treated with retrieval and mouse treated with retrieval plus $CO_2$. Right, AMPAR rectification index ($I_{-80}$ mV / $I_{+60}$ mV). Data are from wild-type (**A**) and $Asic1a^{-/-}$ (**B**) mice. Data are mean±SEM. n = 14–19 for each group. There were no statistically significant differences between groups by ANOVA.

## Inhaling $CO_2$ increases retrieval-dependent activity in neurons bearing the memory trace

To further test if $CO_2$ enhances retrieval-dependent activity in neurons associated with the memory trace, we used the activity-dependent *Fos* promoter to drive expression of GFP variants and mark activated neurons (*Figure 7A*, *Figure 7—figure supplement 1*) (*Reijmers et al., 2007*; *Ramirez et al., 2013*). Aversive conditioning labeled amygdala neurons with long-lasting mCherry and with a short-half life (2 hr) nuclear-localized EGFP (shEGFP) (*Figure 7A*). Immediately after aversive conditioning, mice began receiving doxycycline, which prevents expression of mCherry, but not shEGFP. Twenty-four hours later, we delivered a single retrieval tone or not. Thirty minutes after that, we harvested the amygdala and imaged shEGFP- and mCherry-positive cells (*Figure 7B*).

Compared to controls, inhaling $CO_2$ alone had no effect on the fraction of neurons that were both mCherry-positive and shEGFP-positive (*Figure 7C*). However, a single retrieval tone increased

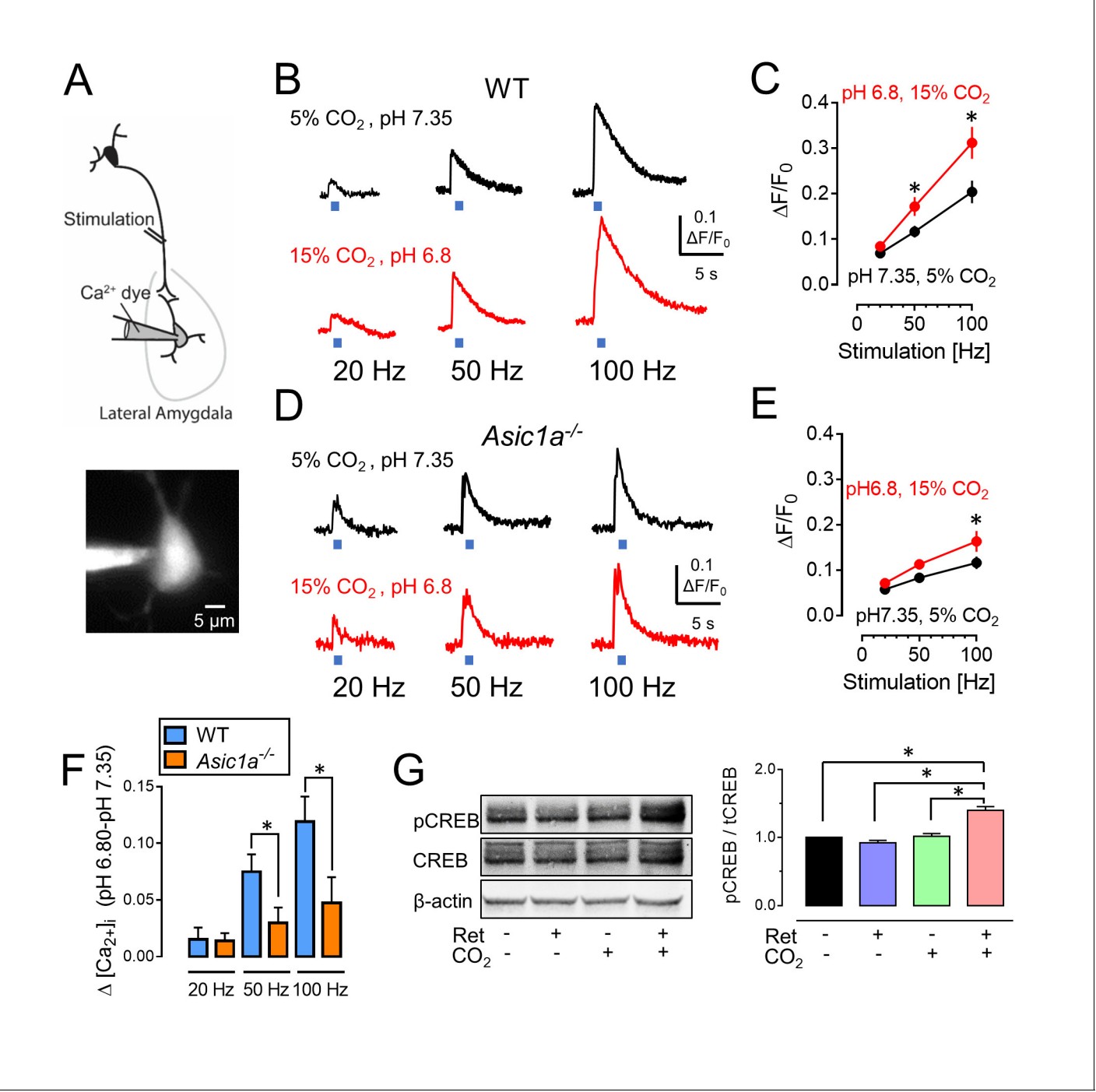

**Figure 6.** $CO_2$-induced acidosis increases stimulation-dependent postsynaptic $[Ca^{2+}]_i$ in amygdala slices and increases CREB phosphorylation after retrieval. (**A**) Schematic for measuring changes in post-synaptic $[Ca^{2+}]_i$. Brain slices were prepared, and lateral amygdala pyramidal neurons were loaded with the fluorescent $Ca^{2+}$ indicator Oregon Green 488 BAPTA-6F (100 μM) via a patch pipet. Changes in postsynaptic $[Ca^{2+}]_i$ induced by presynaptic stimulation at 20, 50, and 100 Hz were assayed when the ACSF was saturated with either 5% $CO_2$ (pH 7.35) or 15% $CO_2$ (pH 6.8). (**B**) Examples of changes in $[Ca^{2+}]_i$ signal with stimulation of wild-type neurons. (**C**) Mean±SEM of changes in $[Ca^{2+}]_i$ signal. *$p<0.05$ by Student's *t*-test. n = 14. p=0.0336 at 50 Hz; p=0.0189 at 100 Hz. (**D,E**) Data as in panels B and C except in *Asic1a*$^{-/-}$ mice. *$p<0.05$ by Student's *t*-test. n = 24. p=0.0215 at 100 Hz. (**F**) Change in $[Ca^{2+}]_i$ signal between pH 7.35 and 6.80 from panels C and E. (**G**) The procedure was the same as that shown in *Figure 3A* except that mice were euthanized 30 min after retrieval. Left, example of western blot with antibodies to CREB phosphorylated on Ser133, total CREB, and -actin. Right, mean±SEM of ratio of Ser133 phosphorylated CREB to total CREB. n = 6 sets of lateral amygdala tissue (each set contained lateral amygdala from the brains of 4 mice). * indicates $p<0.05$ by ANOVA with Tukey's post hoc multiple comparison. No ret vs Ret, p=0.4683; No ret vs No

*Figure 6 continued on next page*

*Figure 6 continued*

ret + $CO_2$, p=0.9819; No ret vs Ret + $CO_2$, p<0.0001; Ret vs No ret + $CO_2$, p=0.2845; Ret vs Ret + $CO_2$, p<0.0001; No ret + $CO_2$ vs Ret + $CO_2$, p<0.0001.

the percentage of mCherry-positive cells that were also shEGFP-positive. When the retrieval cue was presented during $CO_2$ inhalation, even more mCherry-positive cells became shEGFP-positive. These findings indicate that the retrieval cue reactivated neurons bearing the memory trace and that inhaling $CO_2$ augmented the effect of retrieval so that a greater fraction of lateral amygdala neurons showed activity.

# Discussion

These results suggest that when the retrieval tone is presented during $CO_2$ inhalation, an aversive memory becomes more labile than when a retrieval cue is presented by itself. As a result, the memory becomes more amenable to being weakened by extinction or strengthened by reconditioning. It is interesting that although inhaling $CO_2$ reduces pH throughout the brain, it had a specific effect on the response to a retrieval cue. Several findings suggest that inhaling $CO_2$ specifically altered activity in lateral amygdala synapses that were being activated by the retrieval cue and increased the lability of those connections. (1) Inhaling $CO_2$ by itself did not alter AMPAR exchange or the behavioral responses to extinction and reconditioning. (2) $CO_2$ inhalation begun after the retrieval cue did not alter the behavioral response to extinction. (3) Injecting acid alone directly into the amygdala did not increase AMPAR exchange. (4) By itself, $CO_2$ did not increase CREB phosphorylation. (5) $CO_2$ inhalation alone did not increase the percentage of lateral amygdala neurons that were active as reported by *Fos*-driven labeling of neurons that carried the memory trace. Nevertheless, in each of those five cases, simultaneous delivery of $CO_2$ (or acid injection) and retrieval enhanced the effect of retrieval presented alone. (6) Moreover, in contrast to thalamus:lateral amygdala synapses, for cortical:lateral amygdala synapses, where retrieval did not increase AMPA receptor rectification, pairing $CO_2$ with retrieval did not enhance AMPAR exchange.

How did combining $CO_2$ inhalation with retrieval make an aversive memory more labile? In the introduction, we briefly reviewed the evidence that acidosis activation of ASIC currents enhance neurotransmission (*Du et al., 2014*). We speculate that proton:ASIC activity may enhance signaling at synapses while they are activated by the retrieval cue. Neurotransmission can reduce synaptic pH (*Du et al., 2014*; *Highstein et al., 2014*; *Kreple et al., 2014*), and increasing the $CO_2$ concentration will further reduce pH, thereby causing greater ASIC channel activation. ASIC currents may amplify post-synaptic increases in $[Ca^{2+}]_i$ by providing a route for $Ca^{2+}$ permeation or by increasing depolarization (*Ziemann et al., 2009*; *Sherwood et al., 2012*; *Zha et al., 2006*; *Xiong et al., 2004*). That conjecture is supported by our finding that stimulating lateral amygdala input produced a greater increase in $[Ca^{2+}]_i$ when the $CO_2$ concentration was elevated. It is also consistent with the finding of increased CREB phosphorylation (*Sheng et al., 1990*) by the combination of retrieval and $CO_2$ inhalation. It is also supported by the *Fos*-dependent labeling results; adding $CO_2$ inhalation during the retrieval cue further increased the fraction of neurons carrying the memory trace that were activated (shEGFP-labeled).

Previous work on the molecular basis of retrieval emphasized the abrupt and transient exchange from CI-AMPARs to CP-AMPARs in lateral amygdala synapses (*Clem and Huganir, 2010*; *Hong et al., 2013*). Our findings that acidosis further increased AMPAR exchange supports the critical role of CP-AMPARs. The data also suggest that increased activity at those synapses at the time of retrieval was responsible. The abundance of ASIC channels in the amygdala suggests that it may be particularly susceptible to this activity. However, we speculate that synapses in other brain regions might exhibit similar responses and that other behaviors, for example appetitive conditioning, might also be modified.

Translating knowledge about memory modification to humans with disease has so far achieved only limited success. Most studies have focused on behavioral and pharmacological interventions during the labile period that follows retrieval. There has been comparatively less emphasis on strategies to increase the lability of a retrieved memory, although the timing and context for retrieval have

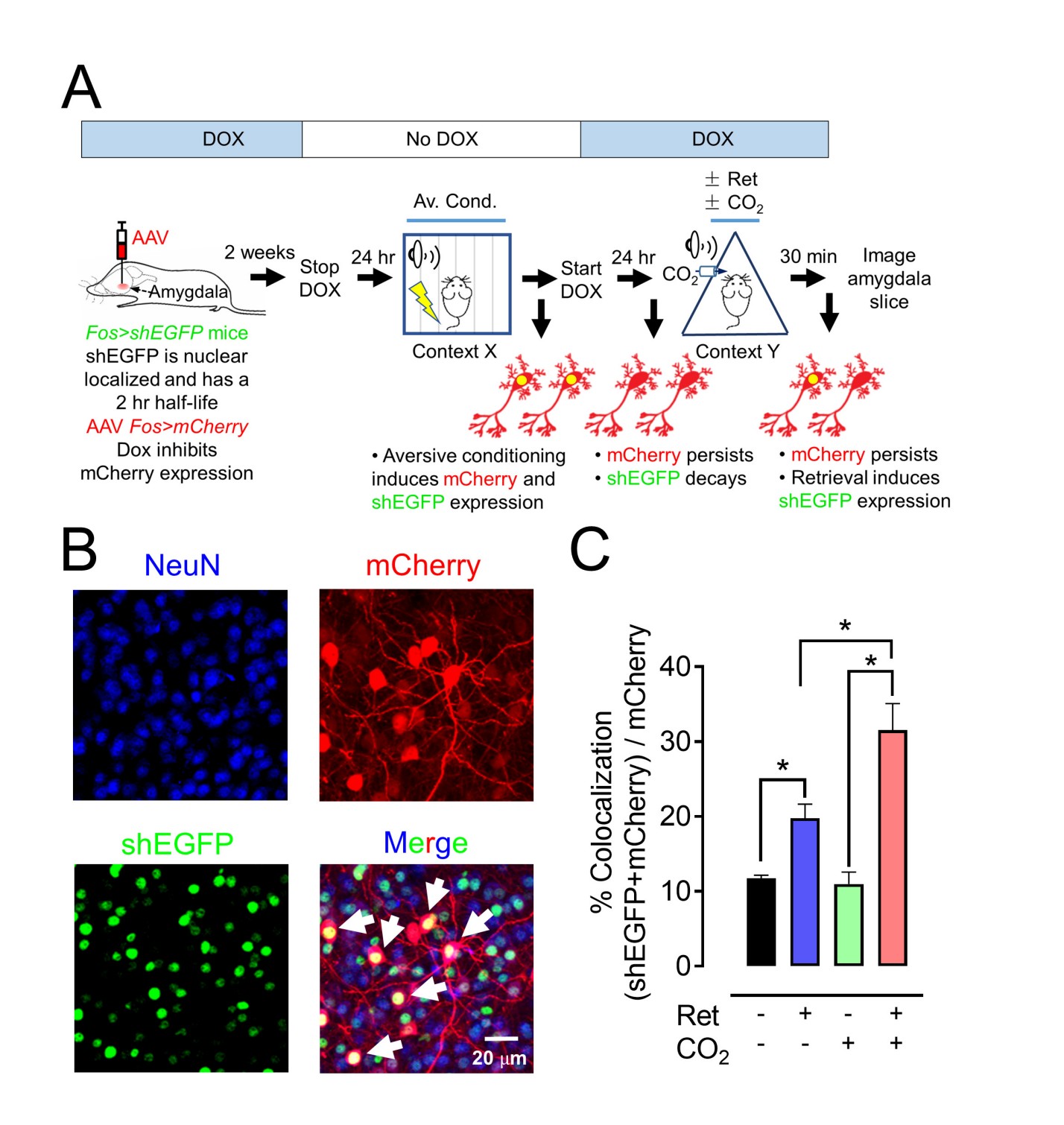

**Figure 7.** $CO_2$ inhalation enhances retrieval-induced activation of lateral amygdala neurons bearing an aversive memory trace. (**A**) Schematic showing procedure; see also *Figure 7—figure supplement 1* for additional information about the doxycycline-inducible system. The activity-dependent promoter *Fos* was used to induce expression of nuclear localized shEGFP in transgenic mice, indicated as *Fos>shEGFP*. shEGFP rapidly decays with a half-life of 2 hr. Mice were fed doxycycline (DOX) for 1 week, and then an AAV9 vector encoding mCherry, indicated as *Fos>mCherry* (which is ultimately expressed when the *Fos* promoter is activated), was microinjected into the amygdala. DOX inhibits mCherry expression. Two weeks later, DOX was discontinued, and mice underwent the aversive conditioning protocol as described in *Figure 1A*. Under those conditions, the *Fos* promoter will drive both shEGFP and mCherry expression, but shEGFP will rapidly decay. After aversive conditioning, mice immediately resumed DOX treatment.

*Figure 7 continued*

One day later, the mice underwent the Ret/$CO_2$ protocol shown in *Figure 1A*. Thirty minutes after that, brain slices were prepared and the shEGFP- and mCherry-positive neurons in the lateral amygdala were determined. (B) Example of lateral amygdala neurons labeled by mCherry (red), the nuclear localized shEGFP (green), and NeuN (blue). White arrows indicate examples of colocalization of mCherry and shEGFP. mCherry- and shEGFP-positive cells were also NeuN-positive. (C) Data are the percentage of mCherry-positive cells that were also shEGFP-positive. Data are mean ±SEM. n = 5 mice for each group. * indicates $p < 0.05$ by ANOVA with Tukey's post hoc multiple comparison. In C, No ret vs Ret, p=0.0154; No ret vs No ret + $CO_2$, p=0.9915; No ret vs Ret + $CO_2$, p<0.0001; Ret vs No ret + $CO_2$, p=0.0197; Ret vs Ret + $CO_2$, p=0.0013; No ret + $CO_2$ vs Ret + $CO_2$, p<0.0001.

The following figure supplement is available for figure 7:

**Figure supplement 1.** Schematic showing how neurons were labeled in mice transgenic for *TetTag Fos-tTA* and microinjected with the viral vector *AAV$_9$-TRE-mCherry*.

been investigated (*Jarome et al., 2012*, *2015*). Our results raise the question of whether $CO_2$ inhalation could be used to increase the lability of a fear memory in humans. Brief $CO_2$ inhalation is a rapidly reversible and safe test used in psychology and psychiatry (*Poma et al., 2005*; *Bailey et al., 2005*). If a means to more effectively make memories labile could be developed, it might have potential for improving treatment of people with post-traumatic stress disorder or phobias (*Schiller et al., 2010*; *Agren et al., 2012*; *Liu et al., 2014*; *Xue et al., 2012*).

## Materials and methods

### Mice

All mice were maintained on a congenic C57BL/6 background. *Asic1a$^{-/-}$* mice (RRID:MGI:2654038) were described previously (*Wemmie et al., 2002*). Experimental groups were male mice matched for age (ranging from 9 to 12 weeks) and assigned randomly to experimental groups. The *TetTag Fos-tTA* mice (RRID:IMSR_JAX:018306) were obtained from Jackson Laboratory (*Reijmers et al., 2007*). They were crossed with C57BL/6J mice (RRID:IMSR_JAX:000664) and selected for those carrying the *Fos-tTA* transgene. The *Fos-tTA* mice have *Fos* promoter driving expression of nuclear-localized, 2 hr half-life EGFP (shEGFP). In these mice, the *Fos* promoter also drives expression of the tetracycline transactivator (tTA), which can bind to the tetracycline-responsive element (*TRE*) site on an injected *AAV$_9$-TRE-mCherry* virus, resulting in the expression of mCherry (*Ramirez et al., 2013*). Doxycycline (DOX) inhibits binding of the tTA to the *TRE* site, preventing target gene expression. We generated *Fos-tTA;Asic1a$^{-/-}$* double transgenic mice by crossing the two transgenic lines. Mice were maintained on a standard 12 hr light-dark cycle and received standard chow and water ad libitum. Animal care and procedures met National Institutes of Health standards. The University of Iowa Animal Care and Use Committee (ACURF #4041016) and University of Toledo Institutional Animal Care and Use Committee (Protocol #108791) approved all procedures.

### Aversive conditioning, memory modification, and memory tests

There were three components to the behavioral tests. The various contexts are in *Figure 1—source data 1*.

#### Auditory aversive conditioning

For all studies, mice were handled for 30 min on each of 2 days prior to aversive conditioning. On day 1, mice were habituated to an infrared aversive conditioning chamber (Med Associates Inc.) in context X for 9 min (*Clem and Huganir, 2010*). Then, the mice were presented with six pure tones (80 dB, 2 KHz, 20 s each). During the last 2 s of the tone, they received a foot shock (0.7 mA, 2 s). The interval between tones was 100 s. They were then returned to their home cage. For experiments in which reconditioning was used for memory modification, mice received 3 tone/foot shock pairs.

#### Memory modification

On day 2, the procedures to modify memories involved two stages. In the first stage, mice were placed in a modified chamber (context Y) and habituated for 4 min Then mice inhaled air or air

containing 10% $CO_2$ for 7 min. Five minutes after initiating inhalation of $CO_2$ or air, mice were presented with one 20 s retrieval tone or not (as a control). One hundred seconds later, they were returned to their home cage. In some experiments, mice received 5% or 20% $CO_2$ instead of 10% $CO_2$. In some experiments, mice were presented with one retrieval tone, and then they breathed the 10% $CO_2$.

The second stage of memory modification occurred 30 min later in context Y. For experiments that involved an extinction protocol, mice received two sets of 20 tones (80 dB, 2 KHz, 20 s each) with an interval between tones of 100 s; for mice that had been presented with one retrieval tone, the first set included 19 rather than 20 tones. Mice were then returned to their home cage. For experiments that involved a memory reconditioning protocol, mice were placed in a different chamber (context Z). After 9 min, they received one tone and foot shock. They were then returned to their home cage.

### Memory test

For mice that had an extinction protocol on day 2, we did two tests on day 7. We tested 'spontaneous recovery' by placing mice in context Y. They were presented with 4 tones (80 dB, 2 KHz, 20 s each, 100 s interval between tones). We tested 'renewal' 30 min later by placing mice in context X and presenting 4 tones (*Clem and Huganir, 2010*). For some experiments, we tested spontaneous recovery and renewal on day 3. For mice that had a reconditioning protocol on day 2, on day 3 we placed them in a different chamber (context ZZ). They were presented with 4 tones.

For each of these interventions, the percentage of time freezing during tone presentation was scored automatically using VideoFreeze software (Med Associates Inc. RRID:SCR_014574).

## Brain slice preparation and patch-clamp recording of amygdala neurons

Ten minutes after memory presentation of the retrieval tone or not, with or without $CO_2$ inhalation, mice were anesthetized with isoflurane and brains were dissected into pre-oxygenated (5% $CO_2$ and 95% $O_2$) ice-cold high sucrose dissection solution containing (in mM): 205 sucrose, 5 KCl, 1.25 $NaH_2PO_4$, 5 $MgSO_4$, 26 $NaHCO_3$, 1 $CaCl_2$, and 25 glucose (*Du et al., 2014*). A vibratome sliced brains coronally into 300 μm sections that were maintained in normal artificial cerebrospinal fluid (ACSF) containing (in mM): 115 NaCl, 2.5 KCl, 2 $CaCl_2$, 1 $MgCl_2$, 1.25 $NaH_2PO_4$, 11 glucose, 25 $NaHCO_3$ bubbled with 95% $O_2$/5% $CO_2$, pH 7.35 at 22–25°C. Slices were incubated in the solution at least 1 hr before recording. For experiments, individual slices were transferred to a submersion-recording chamber and were continuously perfused with the 5% $CO_2$/95% $O_2$ solution (~3.0 ml/min) at 22–25°C. Slices were visualized with infrared optics using an upright microscope equipped with differential interference contrast optics (Nikon ECLIPSE FN1, RRID:SCR_014995).

Whole-cell patch-clamp recordings were made from pyramidal neurons in the lateral amygdala. The pipette solution containing (in mM): 135 $KSO_3CH_3$, 5 NaCl, 10 HEPES, 4 MgATP, 0.3 $Na_3GTP$, 0.5 K-EGTA (mOsm = 290, adjusted to pH 7.25 with KOH). The pipette resistance (measured in the bath solution) was 3–5 MΩ. High-resistance (>1 GΩ) seals were formed in voltage-clamp mode. Picrotoxin (100 μM) was added to the ACSF throughout the recordings to yield excitatory responses. In AMPAR current rectification experiments, we applied D-APV (100 μM) to block NMDAR-conducted EPSCs. To determine current rectification, the peak amplitude of ESPCs was measured ranging from −80 mV to +60 mV in 20 mV steps. Rectification index was measured as the ratio between peak amplitude of EPSCs at −80 mV and +60 mV. In EPSC ratio experiments, neurons were held at −80 mV to record AMPAR-EPSCs and were held at +60 mV to record NMDAR-EPSCs. To determine AMPAR-to-NMDAR ratio, peak amplitude of ESPCs at −80 mV were measured as AMPAR-currents, and peak amplitude of EPSCs at +60 mV at 70 ms after onset were measured as NMDAR-currents. To identify the NASPM-sensitive EPSCs, neurons were held at −80 mV in the presence of 100 μM D-APV. NASPM (100 μM) was applied to the slices during recording. To identify the effects of acid injection on AMPAR current rectification, we implanted a 5 mm 23G guide cannula (Plastics One) above the amygdala. The coordinates relative to bregma were: −1.2 mm anteroposterior; ±3.5 mm mediolateral; −4.3 mm dorsoventral from the skull surface. Mice were allowed to recover for a minimum of 5 days following cannula implantation. On the day of behavioral testing, a 30G injector extending 1 mm beyond the tip of the guide cannula was inserted and 1 μl of saline (pH 3.0 or 7.35) was injected into the amygdala. The injecting solutions were (in mM) 140 NaCl, 2.5 KCl, 2 $CaCl_2$, 1

MgCl$_2$, 1.25 NaH$_2$PO$_4$, 10 HEPES, and 11 glucose. For the pH 3.0 solution we used 10 mM MES instead of 10 mM HEPES as the buffer. Five minutes after acid injections, a single tone was presented to retrieve memory, and 10 min after retrieval, brain slices were isolated for patch-clamp recording. Miniature EPSCs (mEPSCs) were collected at −80 mV holding potential in the presence of 1 µM tetrodotoxin (TTX) and 100 µM picrotoxin. Data were acquired at 10 kHz using Multiclamp 700B and pClamp 10.1 (RRID:SCR_011323). The mEPSCs events (>5 pA) were analyzed in Mini analysis 6.0 (RRID:SCR_002184). The decay time (τ) of mEPSCs was fitted to an exponential using Clampfit 10.1.

## Viral construct and stereotactic injection

The pAAV-*TRE-mCherry* plasmid was obtained from the laboratory of Dr. Susumu Tonegawa (*Ramirez et al., 2013*). The plasmid was used to produce adeno-associated virus (AAV$_9$) by the University of Iowa Gene Transfer Vector Core. The *TetTag Fos-tTA* mice were fed with food containing 40 mg/kg Dox for at least 1 week before virus microinjection. Virus (0.5 µl of 1.45E12 viral genomes/ml of AAV$_9$-*TRE-mCherry*) was injected into the amygdala bilaterally (relative to bregma: −1.2 mm anteroposterior; ±3.5 mm mediolateral; −4.3 mm dorsoventral) using a 10 µl Hamilton microsyringe and a WPI microsyringe pump as described previously (*Ziemann et al., 2009*). After surgery, mice were housed for 2 weeks with DOX-containing diet until behavioral testing.

## Immunohistochemistry and cell counting

Memory trace labeling and analyses are described in more detail at Bio-protocol (*Du and Koffman, 2017*). Twenty-four hours before the *TetTag Fos-tTA* mice injected with AAV$_9$-*TRE-ChR2-mCherry* underwent aversive conditioning, the DOX containing diet was replaced by a regular diet. Immediately after aversive conditioning, the Dox-containing diet was restarted. One day later, mice were presented with the retrieval tone or not, with or without CO$_2$ inhalation. Thirty minutes after retrieval, the mice were euthanized with ketamine/xylazine, and whole brains were fixed through transcardial perfusion with 4% paraformaldehyde (PFA) followed by continued fixation in 4% PFA at 4°C for 24 hr. Then, 50 µm amygdala coronal slices were dissected using a vibratome and collected in ice-cold PBS. For immunostaining, slices were placed in Superblock solution (ThermoFisher Scientific) plus 0.2% Triton X-100 for 1 hr and then incubated with primary antibodies (1:1000 dilution) at 4°C for 24 hr (*Liu et al., 2012*). Primary antibodies include: rabbit polyclonal IgG anti-RFP (Rockland Cat# 600-401-379 RRID:AB_2209751); chicken IgY anti-GFP (Thermo Fisher Scientific Cat# A10262 RRID:AB_2534023) and mouse anti-NeuN (Millipore Cat# MAB377X RRID:AB_2149209). Slices were then washed and incubated with secondary antibodies (Alexa Fluor 488 goat anti-chicken IgG (H+L) (Molecular Probes Cat# A-11039 also A11039 RRID:AB_142924); Alexa Fluor 568 goat anti-rabbit IgG (H+L) (Molecular Probes Cat# A-21429 also A21429 RRID:AB_141761); Alexa Fluor 647 goat anti-mouse IgG (H+L) (Thermo Fisher Scientific Cat# A-21235 RRID:AB_2535804), 1:200 dilution) for 1 hr. Slices were mounted with VectaShield H-1500 (Vector Laboratories Cat# H-1500 RRID:AB_2336788) and viewed using confocal microscopy. We counted mCherry-positive and shEGFP-positive neurons from six coronal amygdala slices (−1.54 mm to −2.34 mm anterioposterior) for each mouse. Reader was blinded to treatment. Co-localization of shEGFP and mCherry was analyzed by ImageJ (RRID:SCR_003070).

## Western blots

Mice were euthanized by isoflurane inhalation 30 min following retrieval or not and CO$_2$ inhalation or not. Brains were isolated, frozen in dry ice, and sectioned on a cryostat (CM 1900; Leica, Bannockburn, IL). Amygdala regions (bregma between −1.6 and −2.3 mm) were punched using a sample corer (1 mm, Stoelting, Wood Dale, IL). Whole cell lysates were isolated from punched tissue as described previously (*Jarome et al., 2012*). Tissue was homogenized, centrifuged at 4000 rpm for 20 min, and the supernatant was collected. Following analysis of protein concentration using the bicinchoninic acid assay kit (Thermo Fisher Scientific, Rockford, IL.), protein samples (40 µg) were separated on 10% polyacrylamide-sodium dodecyl sulfate (SDS) gels (Criterion Tris-HCL precast gel: Bio-Rad, Hercules, CA) and analyzed by western blotting.

Western blotting was performed as described previously (*Price et al., 2000*). The following primary antibodies were used: rabbit-anti-P-CREB S133 (1:300; PhosphoSolutions Cat# p1010-133,

RRID:AB_2492066), rabbit anti-CREB (1:500; Abcam Cat# 1496–1 RRID:AB_562092), mouse anti-$\beta$actin (1:1400, Sigma-Aldrich Cat# A2228 RRID:AB_476697). The following secondary antibodies were used: donkey anti-rabbit IgG IRDye 800CW (1:10,000; LI-COR Biosciences Cat# 926–32213 RRID: AB_621848) and donkey anti-mouse IgG IRDye 680RD (1:10,000; LI-COR Biosciences Cat# 926–68072 RRID:AB_10953628). Membranes were initially incubated with anti-P-CREB antibodies. Following analysis, the P-CREB signal was removed by incubating the membranes in NewBlot PVDF Stripping Buffer (LI-COR Inc.) and reprobed with anti-CREB antibodies. Antibody binding to membranes was visualized using an Odyssey infrared imaging system (Li-COR Inc.). The phosphorylation levels of CREB were determined by normalizing levels of phosphorylated-CREB to the total amount of CREB.

## Confocal $[Ca^{2+}]_i$ imaging

Acutely isolated amygdala brain slices were prepared as described above (Brain slice preparation and patch-clamp recording of amygdala neurons) and used for measurement of $[Ca^{2+}]_i$. Lateral amygdala pyramidal neurons were filled via the patch pipette with Oregon Green 488 BAPTA-6F (100 μM) (Invitrogen, Grand Island, NY) (*Ge et al., 2006*). The pipette solution contained (in mM): 135 $KSO_3CH_3$, 5 KCl, 10 HEPES, 4 MgATP, 0.3 $Na_3GTP$, 10 phosphocreatine (mOsm = 290, pH adjusted to 7.25 with KOH). After obtaining the whole-cell configuration, 15–20 min were allowed for intracellular diffusion of the Oregon green. Cells were then current-clamped for $Ca^{2+}$ imaging. Imaging was performed using a high-speed confocal laser scanning microscope NIKON ECLIPSE FN1 with a long-working distance water-immersion objective (25X; NA 1.1). $[Ca^{2+}]_i$ signals in dendrites near the soma of pyramidal neurons in the lateral amygdala were evoked by electrical stimulation at thalamic inputs with a series of frequencies (20 Hz, 50 Hz and 100 Hz for 1 s). Fluorescence signals were sampled at one frame/10–50 ms. Relative changes in fluorescence were calculated and normalized to baseline measurements as $\Delta F/F_0$, where $F_0$ is the fluorescence intensity before stimulation and $\Delta F$ is the change in fluorescence during presynaptic stimulation. To test the effects of pH on $Ca^{2+}$ influx, the slice perfusate was switched from 5% $CO_2$ saturated ACSF (pH 7.35) to 15% $CO_2$ saturated ACSF (pH 6.8) for 15 min before the $Ca^{2+}$ signal was recorded in response to presynaptic stimulation.

## Statistical analysis

Statistical data was analyzed using Graphpad Prism 6 (RRID:SCR_002798). Data are presented as means ± SEM. Statistical comparison of groups used one-way ANOVA and Tukey's post-hoc multiple comparison test. An unpaired Student's *t*-test was used when only two groups were compared. $p<0.05$ was considered statistically significant.

Sample sizes (n) are indicated in the figure legends. Data are reported as biological replicates, that is, data from different mice, different brain slices, or in the case of western blots, different groups (each group contained tissues pooled from 4 mice). We did not exclude potential outliers. We designed studies with numbers of animals based on our previous experience. For example, because of variable behavior within groups, we used sample sizes of 16–24 mice per experimental group as we previously described (*Price et al., 2014*). In behavioral studies, we typically studied all the groups (often 4 groups) with four animals in each group; then the experiments were repeated with another set of four animals per group, until we reached the target number. Experiments were repeated at the same time of day and with similar handling, habituation and processes. In all cases, animals and samples were randomly assigned to experimental groups.

## Acknowledgements

We thank Thomas Moninger, Jacob Kundert, Rachel Hedinger, Theresa Mayhew, and Sarah Horgen for assistance. We thank Drs. Christopher Benson, Peter Snyder, Leah Reznikov, Yuan Lu, and Collin Kreple for discussions and comments. We thank Drs. Susumu Tonegawa and Xu Liu for providing the TRE-mCherry plasmid. JD is supported by the American Heart Association (15SDG25700054). JAW receives support from the Department of Veterans Affairs (Merit Award), National Institutes of Mental Health (5R01MH085724), and a NARSAD Independent Investigator Award. MJW is an Investigator of the HHMI.

## Additional information

### Funding

| Funder | Author |
| --- | --- |
| American Heart Association | Jianyang Du |
| U.S. Department of Veterans Affairs | John A Wemmie |
| National Institute of Mental Health | John A Wemmie |
| Howard Hughes Medical Institute | Michael J Welsh |

The funders had no role in study design, data collection and interpretation, or the decision to submit the work for publication.

### Author contributions

JD, Conceptualization, Formal analysis, Investigation, Writing—original draft, Writing—review and editing; MPP, Conceptualization, Formal analysis, Investigation, Writing—review and editing; RJT, DG, JJA, ACS, Investigation, Writing—review and editing; MZHS, KS, JM, Investigation; JAW, Conceptualization, Formal analysis, Writing—review and editing; MJW, Conceptualization, Formal analysis, Funding acquisition, Writing—original draft, Writing—review and editing

### Author ORCIDs

Jianyang Du, http://orcid.org/0000-0002-4342-0975
Michael J Welsh, http://orcid.org/0000-0002-1646-6206

### Ethics

Animal experimentation: Animal care and procedures met National Institutes of Health standards. The University of Iowa Animal Care and Use Committee (ACURF #4041016) and University of Toledo Institutional Animal Care and Use Committee (Protocol #108791) approved all procedures.

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
