## [Decision Letter]

Thank you for submitting your article "Transient Acidosis while Retrieving a Fear-Related Memory Enhances Its Lability" for consideration by *eLife*. Your article has been reviewed by two peer reviewers, and the evaluation has been overseen by a Reviewing Editor and Gary Westbrook as the Senior Editor. The reviewers have elected to remain anonymous.

The reviewers have discussed the reviews with one another and the Reviewing Editor has drafted this decision to help you prepare a revised submission.

Summary:

These experiments explore the effect of brain acidification on modification of aversive memories in mice. The work provides evidence that carbon dioxide inhalation during presentation of retrieval cues makes memories more labile. The pathways implicated in the brain involve ASIC1 channels in the lateral amygdala that drive a conversion of excitatory synapses to express a higher proportion of Ca-permeable AMPARs, which make synapses from thalamus more subject to further plasticity, including extinction.

Essential revisions:

The reviewers were favorable about the manuscript, and described the hypothesis as "interesting, novel and [of] possible clinical relevance," the experiments as "well executed and controlled," and the conclusions as "appear[ing] well in line with the results of the experiments." One reviewer summarized the work as follows: "Overall the manuscript is well written, well presented, and elucidates an exciting (and potentially therapeutic) phenomenon related to enhancing lability of aversive memories." Nevertheless, there were a number of concerns that will likely require additional experimentation. Most of these comments fit into the category of tying together the in vitro and in vivo data better. These concerns are summarized briefly here, and expanded on below in the reviewers' own words ("Primary comments"), so that you can have the benefit of their full comments. These include:

1) Testing the role of ASIC1 pharmacologically in wild-type animals, to control for the baseline behavioral changes in the KO mice);

2) Gathering behavioral/imaging/biochemical data from the ASIC1-/- mice and with the Tat-GluA2-3Y peptide. An additional point involves;

3) Clarification of the final experiment, in which the results with the KO cast doubt on the interpretation of data in the final figure. If this cannot be explained or resolved, the experiments could be deleted.

Primary comments:

1) The experiment that could make the manuscript more complete would be with the use of PcTx1 in wild type animals to better assess ASIC involvement as the ASIC1 knockout mice have deficiencies in acquisition of fear related memories, which complicate interpretations of some results.

2) The behavioral data presented is interesting and supports the claim that CO2 present concurrently with the recall cue is required for the facilitation of the subsequent extinction or re-conditioning. Weaker however, is evidence of a common mechanism between the behavioral and slice experiments. Although the authors show that the Asic1a-/- mice were impaired in the reconditioning protocol (Figure 2) At a minimum, the authors should demonstrate that a similar deficit could be observed with the extinction protocol present in Figure 1. This would support their claims of clinical relevance in the Introduction and Discussion. The same general criticism extends to the lack of behavioral testing with the Tat-GluA2 peptide (Cazakoff and Howland, 2011; Dias et al. 2012) and in the mice with microinjections of acidic saline into the amygdala. A full set of consistent behavioral data across these treatments would greatly strengthen the link between the increased memory lability and the proposed mechanism. Likewise, the Ca++ imaging experiments and tone-induced increase in pCREB should have been performed in the Asic1a-/- mice.

3) The results from the Tet-TAG experiment is a bit confusing. Why does retrieval fail to induce reactivation in the Asic1a-/- mice? The 30 minute post-training window allowed for shEGFP expression seems short – what was the rationale for this time point and would the authors expect more overlap at later time points?

---

## [Author Response]

*Essential revisions:*

The reviewers were favorable about the manuscript, and described the hypothesis as "interesting, novel and [of] possible clinical relevance," the experiments as "well executed and controlled," and the conclusions as "appear[ing] well in line with the results of the experiments." One reviewer summarized the work as follows: "Overall the manuscript is well written, well presented, and elucidates an exciting (and potentially therapeutic) phenomenon related to enhancing lability of aversive memories." Nevertheless, there were a number of concerns that will likely require additional experimentation. Most of these comments fit into the category of tying together the in vitro and in vivo data better. These concerns are summarized briefly here, and expanded on below in the reviewers' own words ("Primary comments"), so that you can have the benefit of their full comments. These include:

*1) Testing the role of ASIC1 pharmacologically in wild-type animals, to control for the baseline behavioral changes in the KO mice);*

As we discuss below, this is not feasible.

*2) Gathering behavioral/imaging/biochemical data from the ASIC1-/- mice and with the Tat-GluA2-3Y peptide;*

We have added new behavior experiments with Tat-GluA2-3Y.

*3) Clarification of the final experiment, in which the results with the KO cast doubt on the interpretation of data in the final figure. If this cannot be explained or resolved, the experiments could be deleted.*

As described below, we now discuss this point.

*Primary comments:*

*1) The experiment that could make the manuscript more complete would be with the use of PcTx1 in wild type animals to better assess ASIC involvement as the ASIC1 knockout mice have deficiencies in acquisition of fear related memories, which complicate interpretations of some results.*

Unfortunately, this is not possible. Whereas loss of the ASIC1a subunit eliminates all ASIC currents (in response to challenges of pH>5) in amygdala neurons, PcTX1 only inhibits ASIC channels that are composed solely of ASIC1a subunits. But most ASIC1a subunits in amygdala neurons are in channels that also contain other ASIC subunits, i.e., they are ASIC1a heteromultimers. Moreover, in other experiments, we found that PcTx1 did not alter aversive conditioning or the response to it. We now make the text clearer that our results refer to ASIC currents and not to the ASIC1a subunit specifically (subsection “CO_2_ enhancement of retrieval-induced lability requires ASICs”).

*2) The behavioral data presented is interesting and supports the claim that CO2 present concurrently with the recall cue is required for the facilitation of the subsequent extinction or re-conditioning. Weaker however, is evidence of a common mechanism between the behavioral and slice experiments. Although the authors show that the Asic1a-/- mice were impaired in the reconditioning protocol (Figure 2) At a minimum, the authors should demonstrate that a similar deficit could be observed with the extinction protocol present in Figure 1. This would support their claims of clinical relevance in the Introduction and Discussion.*

We certainly considered this experiment. However, we cannot meaningfully test the effect of extinction on behavior in ASIC1a-null mice. In wild-type mice the% of time freezing after retrieval is small, but we could see a further reduction when we added CO_2_. However, the% of time freezing in ASIC1a-null mice is smaller, and trying to test for a further reduction becomes problematic. Extraordinarily large numbers of mice would be required to test this and uncertainty could persist. That is one reason we tested loss of ASIC1a in a reconditioning experiment. Said another way, it is much easier to test for a loss of a robust increase than a loss of a very small reduction. The reconditioning experiment has the additional advantage that it uses a different protocol, and fulfills the prediction that if a memory is labile it could be either weakened or strengthened.

The same general criticism extends to the lack of behavioral testing with the Tat-GluA2 peptide (Cazakoff and Howland, 2011; Dias et al. 2012).

As suggested, we tested the effect of the Tat-GluA2 peptide on behavior; we show the data in the fourth paragraph of the subsection “CO_2_ inhalation during retrieval augments the exchange of AMPA receptors” and Figure 3—figure supplement 2. These data show that blocking endocytosis of Ca^2+^ impermeable-AMPARs inhibits the CO_2_ enhanced lability of fear memory. We now also reference the two papers the reviewer noted. The results help tie together the in vitro and in vivo experiments and reveal the mechanisms.

*And in the mice with microinjections of acidic saline into the amygdala. A full set of consistent behavioral data across these treatments would greatly strengthen the link between the increased memory lability and the proposed mechanism.*

We had also considered testing the effect of acid injection on behavior, but elected not to try for three reasons. First, we have another experiment with acid injection that links to Ca^2+^-permeable AMPA receptors and an experiment in slices with CO_2_. In addition, the lack of effects of CO_2_ on ASIC1a-/- mice link CO_2_ and ASICs. Second, this experiment requires cannula implants, amygdala microinjection, control injections, and behavior testing. Thus, it would consume a very large amount of time, much effort, and many animals. Third and most importantly, a positive result in this experiment would be fine, as it would be consistent with the conclusion. But, a negative result would be uninformative because of concern about the ability to target a sufficient amount of the relevant circuit bilaterally with an appropriate pH drop. Thus, we believe this experiment will not be revealing.

*Likewise, the Ca++ imaging experiments and tone-induced increase in pCREB should have been performed in the Asic1a-/- mice.*

We have done the Ca^2+^ imaging experiments in ASIC1a-null mice, as well as in wild-type mice (subsection “CO_2_ enhances activation of amygdala neurons” and Figure 6). While the pCREB experiments do support the Ca^2+^ imaging experiments, they are not critical to the arguments in the manuscript.

*3) The results from the Tet-TAG experiment is a bit confusing. Why does retrieval fail to induce reactivation in the Asic1a-/- mice? The 30 minute post-training window allowed for shEGFP expression seems short – what was the rationale for this time point and would the authors expect more overlap at later time points?*

For the Tet-TAG experiment (Figure 7), we now say that we do not have an explanation for why retrieval alone did not increase colocalization in ASIC1a-null mice (subsection “Inhaling CO_2_ increases retrieval-dependent activity in neurons bearing the memory trace”). The editor noted that we could delete this. However, we would like to retain it because of the comparison between the retrieval and the retrieval+CO_2_ groups in wildtype mice. We used a 30 minute interval after retrieval because c-fos expression increases within minutes to hours. For example, one study found that c-fos expression peaked at 30 or 60 min and was markedly reduced at 120 min post-stress (Cullinan WE et al., 1995). Another study found an increase in mRNA expression 30 min after PTZ-induced seizures in rats (Barros VN et al., 2015). It is possible that we might have observed even more colocalization with a longer interval after retrieval, but we did not test that.